

# Behavioural variability among captive African elephants in the use of the trunk while feeding

Maëlle Lefeuvre[1,2], Patrick Gouat[3], Baptiste Mulot[4], Raphaël Cornette[5] and Emmanuelle Pouydebat[1]

[1] Adaptive Mechanisms and Evolution, CNRS/MNHN MECADEV, Paris, France
[2] Institute of Environmental Sciences, Faculty of Biology, Jagiellonian University, Cracow, Poland
[3] Laboratoire d'Éthologie Expérimentale et Comparée E.A. 4443, Université Sorbonne Paris Nord, Villetaneuse, France
[4] Zooparc de Beauval & Beauval Nature, Saint-Aignan, France
[5] Institut de Systématique, Evolution, Biodiversité (ISYEB), Muséum National d'Histoire Naturelle, Paris, France

## ABSTRACT

The Proboscideans, an order of mammals including elephants, are the largest of the Earth lands animals. One probable consequence of the rapid increase of their body size is the development of the trunk, a multitask highly sensitive organ used in a large repertoire of behaviours. The absence of bones in the trunk allows a substantial degree of freedom for movement in all directions, and this ability could underlie individual-level strategies. We hypothesised a stronger behavioural variability in simple tasks, and a correlation between the employed behaviours and the shape and size of the food. The observations of a captive group of African elephants allowed us to create a complete catalogue of trunk movements in feeding activities. We noted manipulative strategies and impact of food item properties on the performed behaviours. The results show that a given item is manipulated with a small panel of behaviours, and some behaviours are specific to a single shape of items. The study of the five main feeding behaviours emphasises a significant variability between the elephants. Each individual differed from every other individual in the proportion of at least one behaviour, and every behaviour was performed in different proportions by the elephants. Our findings suggest that during their lives elephants develop individual strategies adapted to the manipulated items, which increases their feeding efficiency.

## INTRODUCTION

Manipulation of food and tools has a central role in strategies and survival of several different species (*Sustaita et al., 2013*). Although many mammals have been studied in this regard, the literature on primates dominates this field of research. Primates use their anterior limbs and more specifically hands and fingers to manipulate the objects of their interest. Hand grasping, involved in many daily activities, has been extensively studied in primates which are characterised by the ability to individualise their fingers and thus to

Corresponding author
Maëlle Lefeuvre, maelle.
lefeuvre@doctoral.uj.edu.pl

perform complex grasping and manipulation tasks (*Christel, 1993*; *Christel, Kitzel & Niemitz, 1998*; *Christel & Billard, 2002*; *Crast et al., 2009*; *Jones-Engel & Bard, 1996*; *Pouydebat et al., 2009*, *2011*). Grasping and manipulative strategies vary according to the task and the properties of the objects grasped (*Bardo et al., 2016*, *2017*; *Fabre et al., 2013*; *Peckre et al., 2016*). Indeed, the size and the mobility of the object, or the complexity of the task clearly affect the grasping techniques and the hand preference (*Bardo, Pouydebat & Meunier, 2015*; *Peckre et al., 2019a*; *Pouydebat et al., 2010*; *Reghem et al., 2011*). In addition, inter-individual differences in object manipulation behaviours have been quantified in primates (*Bardo et al., 2016*, *2017*). Finally, species lacking suitable hands are also able to manipulate items with a high precision, such as birds with their beaks (*Brunon et al., 2014*) and elephants with their trunks (*Hart et al., 2001*).

The elephants' trunk is involved in many activities such as feeding and drinking, investigation of the environment, vocalisation and social behaviours, and also in making and using tools (*Haakonsson & Semple, 2009*; *Hart et al., 2001*; *Fowler & Mikota, 2006*; *Plotnik et al., 2011*; *Rasmussen & Munger, 1996*; *Shoshani, 1998*; *Yang et al., 2006*). This organ is usually compared with the hand of primates because of its agility and use in various contexts. The distal part of the trunk has prehensile and sensitive capacities helpful for investigation and manipulation.

Different manipulative strategies can be observed in elephants during feeding, depending mostly on the species and the size of the food items (*Racine, 1980*). The two well-developed fingers of the trunk of the African elephants (*Loxodonta* sp.) allow them to pinch small objects with a high precision (*Hoffmann, Montag & Dominy, 2004*) but gripping can be performed differently, for example by wrapping the whole trunk around the item, which is favoured by Asian elephants (*Elephas maximus*). When the item is large, the elephants are more likely to wrap it, and they usually secure the food with the distal part of their trunk, using a vacuum.

The trunk is formed by the fusion of an extended nose and the upper lip (*Rasmussen & Munger, 1996*), reaching 1.5–2 metres long in adult elephants and mainly composed of muscles (between 100,000 and 150,000, *Shoshani, 1998*; *Yang et al., 2006*). Thanks to those muscles, the elephants are able to extend, bend and twist their trunk in both directions by a right-hand and a left-hand array of the helical muscles (*Kier & Smith, 1985*; *Fowler & Mikota, 2006*; *Shoshani, 1998*; *Yang et al., 2006*). This organisation enables them to hold heavy loads as well as catching very small items with a high precision.

Differences between elephants species have been investigated for some behaviours. However, usage of the trunk is usually integrated in more global behavioural categories. *Adams & Berg (1980)* detailed 21 behaviours of captive African elephants, but only seven of them implied the trunk. These included investigation and manipulation, trunk to mouth and eating. *Fowler & Mikota (2006)* explored mostly the chemosensory behaviours and only some of them involved the trunk such as the trunk tip contacts, pinching, blowing, sucking and drinking. No inter-individual differences were evaluated, and to our knowledge no studies investigated the individual-level strategies in trunk use.

The aim of our study was to make a detailed catalogue of the behaviours involving the use of the trunk in captive African elephants. We qualified the uses of the trunk at the

movement level, taking into account the properties of the manipulated items as well as the individual preferences for the different strategies. We hypothesised that (i) the simpler the task (few constraints, like grasping big items) the stronger are the inter-individual differences. On the contrary for more complex tasks involving high constraints, like gripping small items or performing precise tasks, we expected less inter-individual differences. We also hypothesised that (ii) the strategy to manipulate a specific item depends on its size and shape.

## METHODS

### Animal subjects and housing conditions

This study took place at the ZooParc of Beauval from 1 February to 5 April 2019. Six African elephant of savannah (*Loxodonta africana*) females were observed. They were divided into two groups: group A encompassed Juba (named A1 thereafter, 32 year old), Ashanti (A2, 16 year old), Tana (A3, 32 year old) and M'Kali (A4, 30 year old) whereas Marjorie (B1, 33 year old) and N'Dala (B2, 30 year old) constituted the group B. Juba, Ashanti and Tana originated from Knowsley Safari Park (UK) and arrived at Beauval in 2017. The other three elephants came from the Longleat Zoo (UK) and arrived at Beauval in 2003. N'Dala was blind since one year, having a cataract on one eye and retinal detachment on the other. The two groups were never merged, but inside the building the elephants of one group could hear, touch and smell the elephants of the other group through the bars. Encounters of elephants from the two groups were often organised by the keepers with only some of the individuals, in order to merge the groups into a single one in the future. At the beginning of the observation period only one individual of each group participated to those encounters (N'Dala and M'Kali), then Ashanti had been added. No observations were conducted during those particular periods.

During the cold season, the elephants were housed in the building divided into nine boxes. Group A occupied four boxes of 58.5 m² and one of 307 m² (total of 541 m²), whereas group B occupied four boxes (total of 348 m²). On warm days, the elephants get out for a couple of hours in a cement-flooring area connected with the building. As soon as the weather conditions became more favourable, elephants spent up to 10 h per day into the parklands. In this study we observed elephants only indoor.

Elephants had constant access to food in the following categories: hay, tree branches, vegetables and apples. Hay was distributed in every box, in elevated nets which were filled twice a day, in the morning and in the afternoon. Elephants had access to two to three metres long branches of various European trees (i.e. beech, hornbeam, birch) and bamboo. The branches were cut each day and distributed during the filling of the hay nets, in the amount of at least one branch or two bamboos per elephant. We considered bamboos as branches items in our observations because their shape and size were similar. In the morning the branches were laid on the ground outside the boxes, near the bars, while in the afternoon, after cleaning of the areas, they were fixed at the bars and toy installations and laid in the boxes.

Pieces of seasonal European vegetables (a few cubic centimetres, various species such as beetroot, carrots, celeriac, parsnip, fennel, sweet potato, cabbage, cucumber) were

distributed three times per day: at the beginning of the working day, at the beginning of the afternoon and at the end of the working day. They were scattered inside the boxes and outside along the bars.

Finally, apple slices were distributed one by one to capture the elephants' attention during the doors' opening and elephants' transfer between boxes, five times per day when elephants spent the day indoor. During these interactions, we observed elephants' strategies to manipulate apple slices. In our records, we differentiated them from vegetable pieces, because of their different properties. Apple slices were flatter and almost always distributed on the cement flooring, while vegetable pieces were more cubic and also available on the sand flooring.

We worked on captive animals housed in their building, from a distance of approximately 3 m from the bars of the enclosure. These conditions offered many advantages, assuring individual recognition which is essential when studying individual strategies. The proximity to the elephants allowed precise live observations of the movements. Finally, in captivity, the conditions were more standardised than in the wild, thanks to the stable feeding schedule and invariable properties of food items.

### Ethical note

Observations were made following the rules of the zoo and from the security zone, as far as possible from the bars and the elephants. No interactions of the observer with the animals occurred during observations. Only the keepers had their usual interactions with the elephants (moving them, giving them food, and talking to them).

### Behavioural observations

Our aim was to study the use of the trunk by the elephants during their daily activity. Preliminary observations allowed us to identify 65 behaviours displayed in different contexts: feeding, body care, playing, resting, social behaviours and exploration. We predicted a link between the activity and the specific use of the trunk. We also expected that elephants might differ in their use of the trunk during a given task.

We adapted our observation time to the keepers' schedule, in order to avoid interruptions during behavioural sampling if animals were transferred from indoor to outdoor or if the groups' composition was modified by the isolation of an individual. Observations began after the end of the daily training (at around 10 AM) and lasted during the keepers' presence hours in the building, until 7 PM. No observations occurred between 12:30 PM and 2 PM.

Data was collected using the focal animal sampling (*Altmann, 1974*). We defined a sample as a continuous observation of one individual and a session as a sequence of six samples, one for each elephant. Each sample lasted 15 min and one to three sessions were performed per observation day. Before each session, the sampling order was defined thanks to a simulator of random lists. A total of 92 samples were collected, with 14–17 samples per individual. In the afternoon some elephants had access to the cement-flooring outdoor area, and the conditions of observation were not comparable to

**Table 1  Repertoire of feeding behaviours.** Blow, Search behaviours, Trunk to mouth, Contact to mouth and Drink behaviours were removed from the analysis. Pinch, Grasp, Bundling and Block behaviours are reported in *Racine (1980)*, in which Bundling is called Cradling and vary a little bit in the observed uses.

| Behaviour | Description |
|---|---|
| Pinch | Catch little items between the fingers of the trunk. Can be helped by a breath |
| Side pinch | Catch little items between the fingers of the trunk laying down on one side. Taking the item from the side. Can be helped by a breath |
| Blow | Exhalation around an item, usually to clear lightweight elements like hay or sand |
| Grasp | Wrap the trunk around a big item, which would be difficult to maintain with one finger. Potential side preference |
| Torsion/Pressure | Wrap and torsion of the trunk by creating a pressure point, usually to break a branch |
| Bundling | Compact an item on the ventral part of the trunk once or several times before pinching it and bringing it to the mouth |
| Shake | Shake vigorously a pinch held in the trunk |
| Sweep | Sweep with the side of the trunk to gather items before catching them. Potential side preference |
| Gather | Gather or bring back items with the end of the trunk or just the fingers. Potential side preference |
| Pull | Pull an item to break it or separate from the others |
| Search on the ground | Separate items in a pile with the end of the trunk in order to select and sample only a part |
| Search in height | Separate items in a net in height with the end of the trunk in order to select and sample only a part |
| Search in a box | Separate items in an enrichment box with the end of the trunk in order to select and sample only a part |
| Trunk to mouth | Trunk brought to the mouth without providing or removing items |
| Adjust | Adapt the position of an item going out of the mouth, usually a branch |
| Contact to mouth | Brief touch of the mouth with the end of the trunk |
| Drink | Water suction and releasing into the mouth, trunk high-positioned |
| Bring to mouth | Bring a bite to the mouth in order to eat it |
| Block | Wedge an item between the trunk and a tusk, to keep it from others or to manipulate it |

the usual conditions. Hence, we skipped the observation of individuals outdoor, which explains the variable number of collected samples per individual.

During the observation of an elephant, behaviours involving the trunk were scored continuously as occurrences, and the type of activity (e.g. feeding, body care) and the type of manipulated object were reported. The duration of the activity was recorded but the behaviours were considered as events. In this article we present the results related to the feeding behavioural category (listed in Table 1). Feeding behaviours were defined as every movement related to food or water acquisition, manipulation and consumption, as well as mouth's contacts with the end of the trunk while eating (i.e. Trunk to mouth and Contact to mouth in Table 1).

## Data analysis

As described in the literature, feeding was the main activity of the elephants. They spent 72.7% ± 5.8% (mean ± S.D.) of their time feeding and feeding behaviours corresponded to 77.9% ± 4.7% of all the occurrences of behaviours collected. The repertoire of feeding behaviours encompassed 19 different behaviours. We collected 15,234 occurrences but seven behaviours were rarely (<4 occurrences) or never observed during the sampling period. Consequently, they were not considered in the analyses.

In a first step, we tried to establish a link between the different behaviours and the type of food item manipulated. The six different types of food items were: hay, vegetables, apples, and branches (three sizes). Branches were distributed into three different classes (named b1–b3 thereafter) according to their diameter estimated by eye: b1 < 5 mm; 1 cm < b2 < 2 cm; b3 > 2 cm. For each individual we calculated the number of occurrences of each feeding behaviour according to each type of food. Hay was permanently available to the elephants contrary to the other food items and was over-represented in our data set. To avoid this bias we transformed the number of occurrences of the behaviours into the proportion of occurrences, calculated for each food item and each animal. We obtained a table with 10 columns/behaviours and 36 rows (six individuals × six types of food items, $N = 36$). This data set was submitted to a normed Principal Component Analysis (PCA) to detect links between behaviours considered as variables and food items considered as individuals. We then compared the distribution of the coordinates of the 'individuals' on the first component axis to test whether different foods items differed. A nonparametric ANOVA with General Scores and a Monte Carlo procedure was performed, followed by paired comparisons between each type of food items using a Fisher Pitman test for paired samples (pairing by individuals). Because multiple comparisons were made, the probabilities were corrected using a Holm-Bonferroni procedure (*Holm, 1979*) and *P'* (the corrected *P*) is presented.

Hay was always available to the elephants and most of our data were collected when elephants were manipulating and eating hay (62.2% ± 7.9% of feeding occurrences). The inter-individual differences were tested in comparing the behaviour of the animals when feeding on hay. Because the total number of occurrences varied greatly between animals and samples of a given individual we used the proportions to compare the animals. The comparisons were held only for the five behaviours with enough occurrences. As the two groups did not have exactly the same physical and social environment, we compared the animals of each group separately. All the samples with less than 20 occurrences of behaviour were discarded from the analysis. Consequently the number of samples per animal differs slightly (A1: 12; A2: 10; A3: 11; A4: 9; B1: 14 and B2: 11). For each behaviour, we compared the two individuals of group B with a Fisher Pitman test for independent samples; for the four elephants of group A, we first made an ANOVA with General Scores (with the Monte Carlo procedure) followed by paired comparisons using Fisher Pitman tests for independent samples. The probabilities were corrected using the Holm-Bonferroni procedure and the corrected *P* value (*P'*) is reported.

The PCA was carried out using Statistica software and the ANOVAs and Fisher Pitman tests were made using StaXact software.

## RESULTS

### Manipulation behaviours and food items

The first two components of the PCA explained 55.3 % of the variance in the correlation matrix (detailed results are presented in Supplemental Material). Component 1 opposed two groups of behaviours (Fig. 1). Block, pull, adjust, grasp and torsion are on the right side whereas bring to mouth, gather, side pinch and sweep are on the left side.

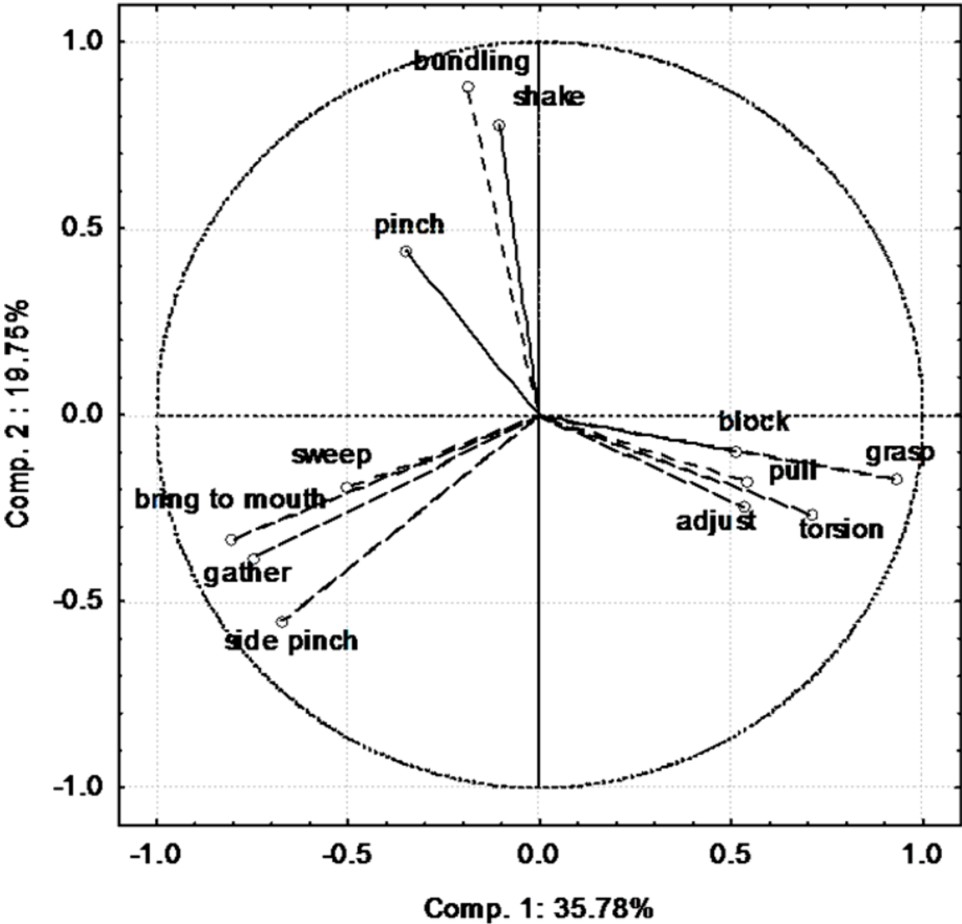

**Figure 1 Contribution of the different variables (behaviours) to the first two components of the PCA.**

Bundling, shake and pinch are poorly represented on component 1 and were mainly represented on component 2.

The different types of food items are distributed into three separate groups (Fig. 2): (i) hay, (ii) the three sizes of branches and (iii) vegetables and apples. The analysis of the coordinates on the first component confirms this distribution into three groups. The ANOVA with general scores on the total set of data (i.e. the six initial groups) is highly significant ($P < 0.0001$) and a similar result is obtained when food items are divided into the three groups described above ($P < 0.0001$). The three groups differ significantly from each other (branches-hay: $P' = 0.001$; branches-vegetable and apple: $P' < 0.001$; vegetable and apple-hay: $P' = 0.003$). There are no pairwise differences between the three sizes of branches (b1–b2: $P' = 0.09$; b1–b3: $P' = 0.18$; b2–b3: $P = 0.81$), nor between vegetables and apples ($P = 0.24$).

## Inter-individual differences in the manipulation of hay

During the manipulation of hay by the elephants, we observed five main behaviours (Fig. 3). These were the three behaviours revealed by the PCA (bundling, shake and pinch; Fig. 1) and two others; grasp and bring to mouth.

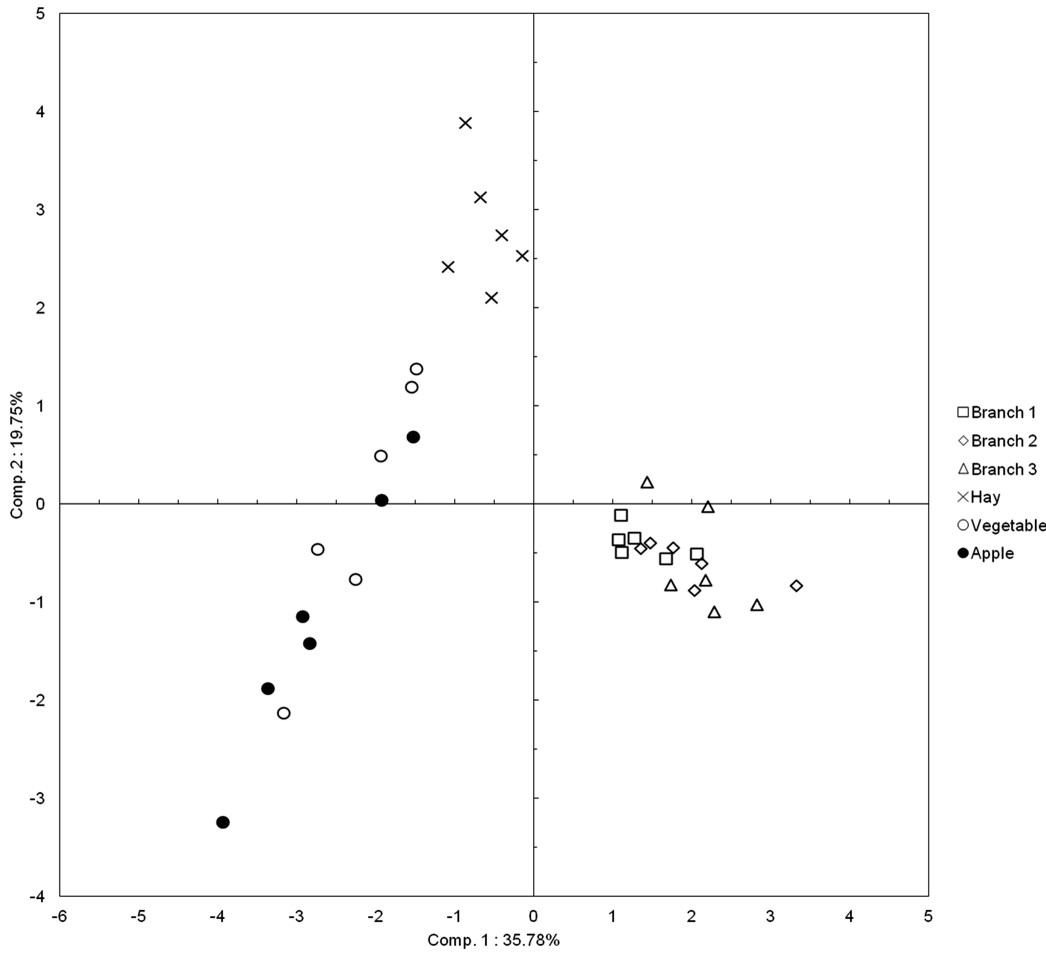

**Figure 2 Plot of the food items in the two first dimensions of the PCA.** Each point of a given type of food item corresponds to a different elephant ($N = 6$).

The use of each of these five behaviours by the elephants was analysed independently and the comparisons between individuals were made separately inside each group (Fig. 4). Pinch is the most common behaviour used by the six elephants. No difference is found in group B ($P = 0.29$), whereas in group A elephants differed in the frequency of using it (Anova with general scores, $P < 0.0001$). M'Kali (A4) used this posture significantly more often than the other three elephants ($P' < 0.012$ whatever the comparison) and Ashanti (A2) used significantly less this posture ($P' < 0.004$ in each pairwise comparison). Juba (A1) and Tana (A3) do not differ significantly ($P = 0.16$) and occupy an intermediate position.

The elephants in group A do not differ significantly in their use of bundling (ANOVA with general scores, $P = 0.06$) and none of the pairwise comparisons is significant ($P' > 0.16$). In group B, the proportion of bundling is higher for Marjorie (B1) than for N'Dala (B2) ($P = 0.026$).

Only the elephants in group A differ significantly in their use of shake (ANOVA with general scores, $P = 0.0001$) but only marginal differences are found in pairwise
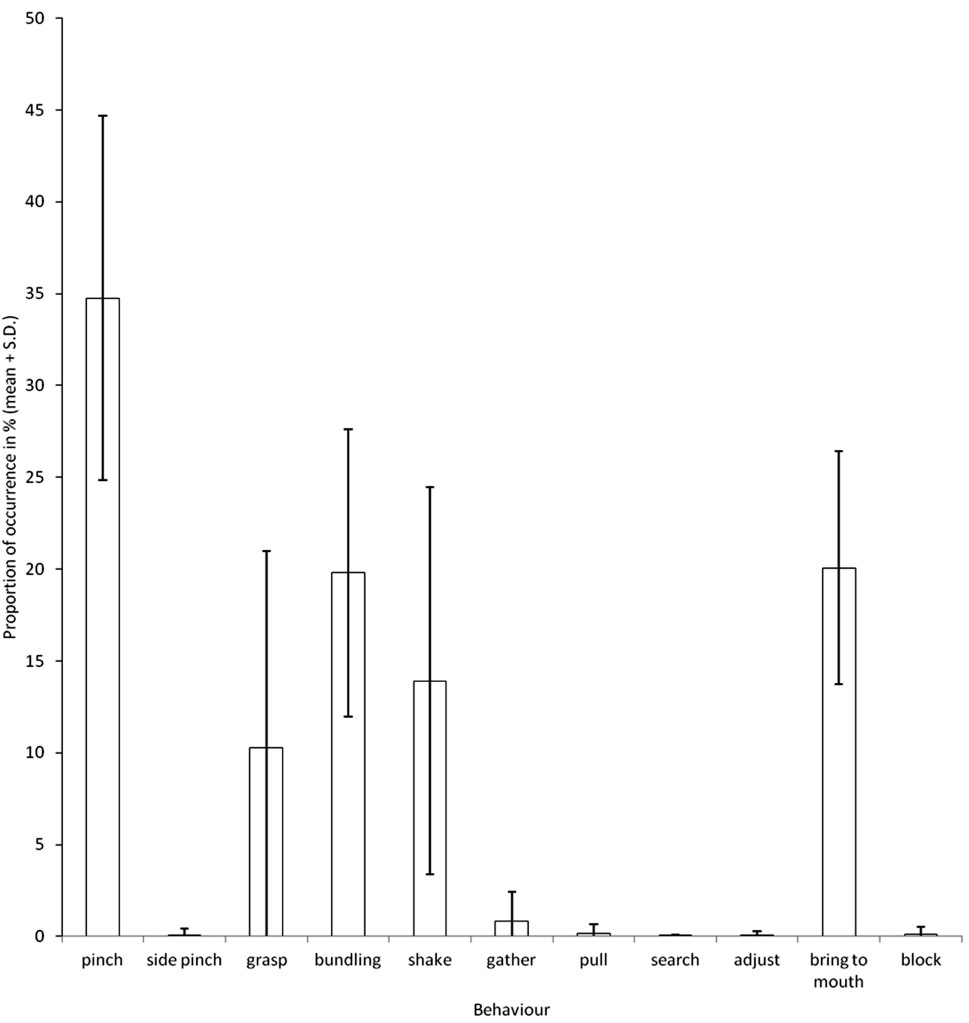

**Figure 3 Proportions of occurrences of the different behaviours observed during feeding.**

comparisons. The proportion is the highest in Tana (A3) but the differences are significant only with Juba (A1) and M'Kali (A4) ($P' = 0.035$ and $P' = 0.01$ respectively). No difference is found in group B ($P = 0.50$).

The elephants in group A differ significantly in their use of grasp (ANOVA with general scores, $P < 0.0001$). The proportion of grasp is the lowest in Juba (A1) and the highest in Ashanti (A2) ($P' = 0.009$) and the two other elephants occupy intermediate positions and no significant difference is found ($P' > 0.08$). In group B, the proportion of grasp is more than 10 times highest in Marjorie (B1) than in N'Dala (B2) ($P < 0.0001$).

The elephants in group A differ significantly in their use of bring to mouth behaviour (ANOVA with general scores, $P < 0.0001$) and this difference is mainly due to the low score of M'Kali (A4) compared to the other three elephants ($P' < 0.0002$ in each pairwise comparison).

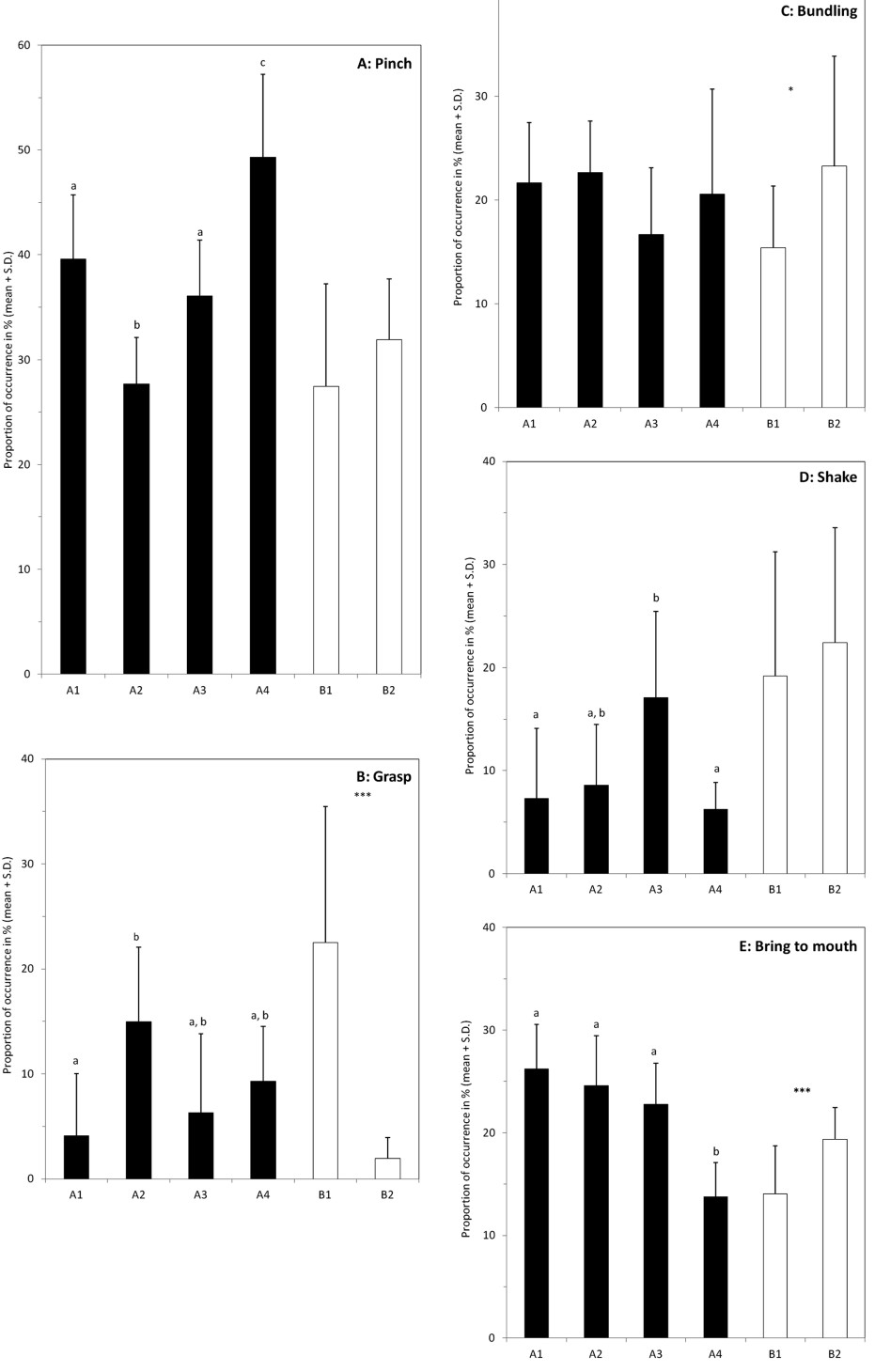

**Figure 4 Individual differences in the use of the different behaviours when feeding on hay.** Graphs (A–E) show proportion of occurrences of the five main behaviours involved in the hay consumption (name at the top right of the box). The black bars correspond to the four elephants of the group A (A1–A4), and the white bars to the two elephants of group B. For group A, bars with no common letter differ significantly; significant differences between the two individuals of group B are indicated by: *($P < 0.05$) or ***($P < 0.001$). The number of samples varies between the individuals (A1: $N = 12$; A2: $N = 10$; A3: $N = 11$; A4: $N = 9$; B1: $N = 14$ and B2: $N = 11$).

## DISCUSSION

Our results revealed a significant inter-individual variability in the usage of the main feeding behaviours to manipulate a restricted range of food items. We observed elephants in an artificial environment with a limited variety of vegetal species compared to their natural habitat. *Buss (1961)* reported 20 browse species found in the stomachs of 79 wild African elephants from savannah in Uganda. Grass was the main food consumed, and leaves, branches and fruits represented 10 percent of the stomach content. He also reported the consumption of succulents and herbaceous plants such as aloe, cotton and papyrus which are not provided in captivity. Although vegetable species were less diversified in the wild than at the zoo, the shape, size and distribution of the pieces were similar for all the types of vegetables. In our study the availability of each food type was well known by the elephants as well as their localisation. The food distribution was designed to minimise competition between animals and under these conditions we expected the elephants to have low constraints to express their own behavioural feeding strategy.

Despite a limited variety of foods, the elephants' diet encompassed different types of items. Hay was the main component and was available ad libitum, branches of different sizes and pieces of vegetables were distributed periodically in the course of the light period whereas slices of apples were specifically given to the elephants during the opening of the doors. Our results revealed an outright link between food items and behaviours. For instance branches were mainly grasped and their manipulation usually entailed torsion to break them. Branches were the only food item implying an adjustment when they were too long and thick to be contained entirely into the mouth. We were unable to find any significant differences in the manipulation of branches according to their diameter, suggesting the relevance of the structure of an item more than its size in the adopted feeding strategy. To our knowledge, the diameter of the branches has never been studied in relation to their manipulation by elephants. Wild African elephants have been observed selecting branches and foliage of tall and frequently used trees (*Smallie & O'Connor, 2000*). This resource hedging shaped the foliage production and increased the short-term food available quantity. Nevertheless, captive elephants have ad libitum resources and rarely feed on living plants. Thus, a hedging strategy has no meaning in captive populations. *Hart et al. (2001)* reported branches modification by wild and captive Asian elephants in order to get rid of flies. They did not mention the diameter, only the length of the branch and the process were reported. To break a side branch, elephants used their foot to maintain the rest of the branch on the ground. The 0.75–2 m-length side branch was used as a tool against flies and then was eaten or dropped. Our study focused on the uses of the trunk, hence we did not report the involvement of feet in the feeding processes. Nonetheless, from our personal observations, it is worth noting that elephants scarcely implied their foot during food manipulation. They preferred to maintain branches with their tusks or in their mouth.

The hay was pinched or wrapped, and it was very often shaken and bundled. Those two behaviours were observed also with other food items but more rarely. Generally, they were undertaken to remove sand or to improve the grip. Surprisingly, pieces of apples and

vegetables were managed in a similar way. Elephants pinched them with the fingers of the trunk, and more rarely gathered and wrapped them, in order to take several pieces at once. These results suggested that the small size of the food item was more determinant than its shape (slice vs piece). Besides, the way of delivery of the food items seemed not to be decisive in the selection of the feeding strategy. *Racine (1980)* also observed the manipulative strategies of captive Asian and African elephants with different kinds of food items (apples, oranges, watermelons). He correlated the different strategies with the trunk morphology (i.e. smaller fingers in Asian than African elephants) and the size of the food. Small items easily maintained in the distal part of the trunk were pinched while other strategies were employed with bigger items, mostly grasping. One alternative strategy, which we did not mention in our behavioural repertoire, was described by *Racine (1980)* as a position referring to 'a golf ball on a tee.' The elephants maintained the item on their opened fingers and then pushed up the food to their mouth. The distance between the elephant and the food might also impact the manipulative strategy, nearer food items being more often grasped than distant items (*Racine, 1980*). In our study, no distance-dependent strategy appeared. Elephants used predominantly the pinch behaviour to grip vegetable pieces either in front of them or outside the enclosure. We observed that grasping movement was more related to the quantity of items than to their distance. Grasping was mainly performed to catch several items at once. One similarity which our observations share with Racine's is the predominance of pinching behaviour to catch food against walls. In this situation, the grasping movement can be constraining, and we argue that pinching is a more efficient strategy. Finally, we expected that substrate characteristics should influence the vegetables feeding strategy. Indeed, grasp movement was exclusively observed on the cement-flooring around the enclosure. Grasping on the sand-flooring would certainly gather more sand than vegetable pieces and dramatically reduce gripping efficiency.

Elephants are not the only animals changing their strategy depending on the item properties. This research domain has been extensively studied in primates, showing how the properties of the food (size, mobility) affected gripping kinematics and general strategies, such as the use of the mouth versus the hands or the various techniques with hands (*Nekaris, 2005*; *Peckre et al., 2019a*, *2019b*; *Petter, 1962*; *Pouydebat et al., 2009*; *Pouydebat & Bardo, 2019*; *Scheumann et al., 2011*; *Toussaint et al., 2013*, *2015*). It is really interesting to demonstrate that the adaptation of gripping techniques is not limited to the hand, but can be addressed in other prehensile organs (*Brunon et al., 2014*; *Sustaita et al., 2013*).

Although the elephants shared common behavioural repertoires, they differed in the proportion of the usage of those common behaviours, at least when they were eating hay. In group A, inter-individual differences emerged in four of the five major behaviours observed (Fig. 4). Only the proportion of bundling was similarly used by the four elephants. It is worth noting that all the elephants compared pairwise differed significantly at least in the use of one behaviour, and pinch was the behaviour with the most important inter-individual variability. Similar results were found within group B. The two elephants used differently the grasp, bundling and bring to mouth behaviours whereas they

performed shake and pinch movements with the same proportion (Fig. 4). Shake was more used by both females in group B than females of group A with the exception of Tana (A3). Yet, despite the two groups were in different enclosures in the same building, there was no clear difference in the quality of the hay or in the pattern of hay distribution that could explain this difference. In our study we reported behaviours as events and we did not consider their duration. In our ethogram, we defined most of the behaviours at the movement level and consequently, they were usually performed quickly. However equal weight was given to short and long movements, which could have inhibited deeper inter-individuals differences.

One of the elephant of group B (N'Dala, elephant B2) became blind one year before the observations and we expected that this female could have developed specific strategies. The difference between the two females in the group B was the highest for grasping, as the blind female displayed distinctly less grasp behaviour than the other. The performances of the blind female did not present explicit differences with the other elephants, and we were unable to show the emergence of a specific strategy when eating hay. Possibly, the sense of sight plays a secondary role in elephant feeding, olfaction and haptic senses may be more important and the trunk is properly equipped. Moreover, since observations were made inside the building, in which the limited space was well known to the elephants and no specific behavioural strategy was required for a blind elephant. Only a comparison with the behaviour of this individual before the occurrence of blindness could have revealed a behavioural modification. In human, brain plastic adaptation to blindness has been extensively investigated. The compensation of the loss of eyesight seems to be generated by complex tasks only (*Gizewski et al., 2003*) but the over-development of the other senses in early- and late-blind people failed to win unanimous support. Opposite results emerged on exacerbated tactile capabilities and olfactory sense (*Sathian & Stilla, 2010*). The outcome appeared to depend on the task and the practice. In addition, a recent large-scale study found no difference between early-, late-blind and sighted people in different olfactory tests (*Sorokowska, 2016*). We consider cautiously those results because of the poor vision of elephants and their far more accurate sense of smell than the one of humans.

Authors studying different species tried to explain the inter-individual differences they noted in their experiments. *Racine (1980)* reported different strategies for the six elephants he observed. He proposed three explanations of variation in individual behaviours: morphology, learning and captivity. He studied both Asian and African elephants, with different shapes of the distal part of the trunk and an effective difference in the use of pinch and grasp behaviours. The trunk morphology of our elephants was less variable, with the exception of the very short ventral finger of Ashanti which recalled an Asian-like trunk. Yet, her feeding strategy was not significantly different from the other individuals, so the morphology cannot be the only factor explaining variation in behaviour. The cohabitation of closely related species from different locations and with different backgrounds can lead to the transmission of behavioural wonts and enlargement of the repertoire (*Galef & Giraldeau, 2001*). African elephants living in captivity with Asian elephants integrated the use of foot in items manipulation (*Racine, 1980*). Elephants are

fast learners (*Plotnik et al., 2011*) and gathering social animals may homogenise their catalogue and frequencies in the usage of feeding behaviours. Yet our results failed to show a group-level similitude. For instance the side pinch behaviour was observed during vegetables and apples feeding but was removed from the analysis because of a low number of occurrences. This behaviour was favoured over the pinch behaviour by some individuals including Ashanti (A2) and Marjorie (B1). They originated from different locations and were housed separately at the zoo. Thus it is probable that the side pinch behaviour appeared independently without transmission between our elephants. We can hypothesise that behavioural learning occurs when a behaviour is more efficient than the previously used strategy. However, we would need further information about their early life to investigate this possibility.

Gripping variability has been studied in mammals as well as in other taxa (see *Sustaita et al., 2013* for a review). Pigeons (*Columba livia*) have been shown to adapt their pecking movement to the food size and accessibility (*Siemann & Delius, 1992*). Opening of their mandibles depended on the size of the seed and the gripping and pulling strength differed between free and attached seeds to the substrate. Recorded movements were highly variable between individuals. This variability was explained mainly by learning as pigeons consume highly diverse food and forage in diverse situations. In our study, we highlighted the influence of the food shape on the feeding strategy, but we raised no evidence of the impact of attached food. Different strategies could emerge in response to more hardly obtainable items. In captivity, the feeding conditions are unidentical to conditions in the wild, where the food is usually attached to trees or substrate. In more naturalistic conditions, elephants could demonstrate a larger repertoire of behaviours aiming the processing of living plants into bite-size items. The availability of food in captivity could inhibit natural behaviours and more pronounced inter-individual differences in food acquisition.

Gripping behaviours play an essential role in locomotion, feeding, and reproduction in a great diversity of tetrapod vertebrates, but has received relatively little attention outside of the anthropological, primatological and biomedical literature (*Sustaita et al., 2013*). Although the ability to reach for food or prey or substrate, to hold it in a forepaw, or manipulate it with the digits is sophisticated in primates, gripping abilities and manipulation can also be highly developed at least in other mammals. Gripping modalities may differ from group-to-group, but they share common muscles bases and selective pressures (*Sustaita et al., 2013*). We need to explore them much more across taxa, outside primates.

## CONCLUSION

Proboscideans are characterised by their big size and especially by their prehensile and sensitive trunk. It is usually compared with the primates' hand, and similarly to this group, elephants show individual strategies. In our study we focused on feeding behaviours and we highlighted the important influence of the food properties on the gripping strategies. The elephants identified the item and adapted their movement to manipulate it efficiently. The same behaviours were entailed to consume pieces of vegetables and apples

or branches of different diameters, leading to the conclusion that the overall shape and size of the food is determinant for the employed strategy. Inter-individual differences in behaviour were especially clear during hay consumption. Despite a similar repertoire, each elephant differed from the others in the frequency of at least one behaviour. Neither blindness of N'Dala nor the particular trunk morphology of Ashanti seemed to be the factors of this variation. Inter-individual behavioural differences could be triggered by early learning or intrinsic preferences. However, further observations are needed to investigate the impact of preferences on feeding strategies and behavioural variability. Gripping performance might play a more critical role in tetrapod evolution than currently understood. More comprehensive data on gripping behaviour and functional morphology, from a greater diversity of taxa, are required to test this in a rigorous phylogenetic framework, and elephants have to be included in these studies.

## ACKNOWLEDGEMENTS

We gratefully thank the ZooParc of Beauval who owns the elephants and welcomed our project. Experiments were conducted with the help of the keepers under the direction of Yann Ménager. All our thanks go to him and Nathan Durand, Amaury Boutier, Matthieu Villemain, Matthieu Fromet, Mathieu Hysbergue, Mégane Marron and Clément Langles.

### Funding

This work was supported by the ATM of The National Museum of Natural History (France) and the funding CNRS 80 PRIME. The funders had no role in study design, data collection and analysis, decision to publish, or preparation of the manuscript.

### Grant Disclosures

The following grant information was disclosed by the authors:
ATM of The National Museum of Natural History (France).
CNRS 80 PRIME.

### Competing Interests

The authors declare that they have no competing interests.

### Author Contributions

- Maëlle Lefeuvre conceived and designed the experiments, performed the experiments, prepared figures and/or tables, authored or reviewed drafts of the paper, and approved the final draft.
- Patrick Gouat conceived and designed the experiments, analysed the data, prepared figures and/or tables, authored or reviewed drafts of the paper, and approved the final draft.
- Baptiste Mulot conceived and designed the experiments, authored or reviewed drafts of the paper, and approved the final draft.

- Raphaël Cornette analysed the data, authored or reviewed drafts of the paper, and approved the final draft.
- Emmanuelle Pouydebat conceived and designed the experiments, authored or reviewed drafts of the paper, and approved the final draft.

## Data Availability

The raw data is available in the Supplemental Files.

## Supplemental Information

Supplemental information for this article can be found online at http://dx.doi.org/10.7717/peerj.9678#supplemental-information.

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
