# Peer review of "Behavioural variability among captive African elephants in the use of the trunk while feeding"

_PeerJ, doi:10.7717/peerj.9678_

## Round 0.1 · original submission · Minor Revisions

Thank you for submitting an interesting manuscript.

The three reviewers highlight the need to make some corrections to the English language to improve clarity and accuracy. Each reviewer has provided examples, annotations and suggestions which will be a great help in this regard.

The methods section for behavioural observations needs to be clearer (again see reviewers' suggestions). You have defined a session but not a sample and the distribution of observations sessions for each elephant across each day is not clear. A small table might be a good way to summarise dates, times and durations of observations. The reviewers are confused as to whether you recorded behaviour only as events (frequency of occurrence) or as states (frequency and duration). Please clarify and explain more clearly how data were transformed to proportions of observation time.

Reviewer 1 highlights the limited range of feed types received by the elephants. You mention this on p239, but this could be given a little more emphasis.

Reviewer 2 asks whether the detailed statistical analysis is useful. I feel that it is helpful in showing that different manipulation behaviours are used in different contexts, so would like you to keep this. I also understand the reasons to keep Group A and B elephants separate (due to different living environments). However, I agree with Reviewer 2 that there are important details about the PCA are missing, so please include these in the revised version.

Please be very careful in your use of terms such as "high variability" or (line 238) "great variation"...… What reference point do you use to conclude that the variation between individuals is high? Figure 4 shows that the elephants all use the same behaviours to a greater or lesser degree. From the title I expected that perhaps some individuals had developed completely unique methods.

The discussion is rather long and, although it is good to highlight the possible influence that blindness or trunk morphology may have had it is not possible to analyse this further. The general discussion on blindness in humans on lines 328-347 is speculative and far removed from this study and should be greatly shortened.

Finally, please could you include some data on the potential side preferences of the elephants. The ethogram lists 4 behaviours where you state that side preferences might arise, and you also refer to the work by Haakonsson - please, therefore, report whether side preferences were observed in any or all of the elephants, and whether there were any correlation between side preference and preferred manipulation methods. This would be relevant to the growing topic examining lateralised behaviour, individual differences and animal welfare in captive environments.

Reviewer 1 ·

Basic reporting

This section is fine.

Experimental design

This also is fine.

Validity of the findings

Findings are accurate but see longer comments to author.

Additional comments

This paper offers detailed observations on styles of eating among a couple small groups of elephants at a zoo. To begin, the paper's title really needs to be more specific and informative, something like, Variability among a zoo's African elephants in use of the trunk while feeding. (It doesn't seem like HIGH behavioral variability, and POPULATION is too sizable a term for this study, plus it pertains only specifically to use of the trunk while feeding. Some variability is to be expected, e.g., individual elephants vary in their interactions with humans, Rossman et al.)

The study is well-executed with extensive detailed observations. It first provides a very nice comparative review on manipulation of food by primates and elephants with uses of hands, fingers, and the elephant's trunk. Then it sets forth the goal of examining how the trunk is used in feeding, with a couple of hypotheses. Elephants were provided items of food each day, the largest being branches 2-3 meters long.

Elephants living in the wild or in naturalistic environments are surrounded by living plants, ranging from grasses available on the ground to huge trees. Using their trunk, tusks, and feet, they are well-equipped with a range of strategies for processing these growing items into food for themselves. A limitation of this study is that it takes place in a stripped-down environment that offers no opportunities for elephants to exhibit their typical approaches for acquiring and preparing food items for consumption.

There is no mention of elephants using the foot to assist in feeding during these observations, perhaps because, with food being delivered in elephant-bite-sized pieces, there was no need to use the range of special skills elephants employ to prepare their food. Using personal communications, I have verified that both wild and domesticated Asian elephants in India use their feet to assist in feeding, not only to break branches into more manageable pieces, but also to knock dirt off of grass clods that are held by the trunk (personal communication, L. Hart, May 12, 2020). These are routine behaviors performed by elephants as they are feeding. And in South Africa, captive African elephants in a spacious naturalistic environment show similar behavior with grass clods, and also use the foot to break up items such as a melon into manageable pieces (personal communication, Z. Rossman, May 10, 2020). The zoo environment in this study provided only limited avenues for manipulating items, or at least there was little need to use a wider range of strategies with the food and environmental items that were provided.

It seems the study revealed the stringent limitations of the zoo environment, resulting in the elephants exhibiting much less of their behavioral repertoire than would be seen in a more naturalistic environment where they are surrounded by growing vegetation. This limitation bears discussion in the paper.

Nonetheless the paper provides some interesting data and i support publishing it.

Several typos are highlighted. Many of these are incorrect singular/plurals, or apostrophes. Also, the format is inconsistent in the references regarding upper and lower case for the titles.

Annotated reviews are not available for download in order to protect the identity of reviewers who chose to remain anonymous.

Reviewer 2 ·

Basic reporting

Abstract
Whilst the opening line of the abstract sounds good, it’s not very accessible to a general audience, or a lay one, and it’s not factually correct as it’s the order Artiodactyla that contains the largest mammals on Earth; the whales. I would suggest explaining which animals you are referring to, for example, “The Proboscideans, an order of mammals including elephants, are the largest of the Earths land animals”.

Introduction
Line 26: change ‘able’ to ‘ability’
Line 32: remove ‘also’
Line 37: add ‘elephants’ before ‘trunk’
Line 43: change ‘individuals’ to ‘elephants’, and merge with the following sentence to read: “…1.5-2 meters long in adult elephants, and is mainly composed…” (I suggest this because grammatically, the second sentence isn’t complete since it starts with ‘it’).
Line 47: a sentence shouldn’t really start with ‘it’.
Line 64: perhaps add something like ‘in trunk use’ at the end. I won’t comment any further on sentence structure, or English language, as that’s not what reviewing is for.

Methods
Line 76: Add the year.
Please make it clear that you did live observations, perhaps in line 117 where you discuss proximity to the elephants. Also, please make clear what this proximity was in some sort of distance estimate.

Behavioural observation: please describe fully how you recorded the behaviour, i.e. I am assuming from what you’ve written that you scored feeding behaviour continuously during each 15-minute focal sample, as you appear to have durations for feeding as a state behaviour, but this hasn’t explicitly been stated nor has a definition of ‘feeding’ been given. I see you have an ethogram with definitions of each type of manipulation behaviour you recorded, but this should be referenced here not in Data Analysis. You also need to state how you recorded these; i.e. were they events counted as frequencies, which you then converted to percentages for each individual per food item? These methods need to be described in enough detail to be replicated.

Line 159: was branch diameter measured before the branches were put in or estimated by eye?

Line 163: number of occurrences of what? This isn’t clear.

Line 169: what is a ‘general score’?

Discussion
The discussion is good, well done. I have a few minor comments.

Line 265 onwards: It is not clear here if you are still referring to the Hart study or back to your own data again.

Line 324: ‘may be more important’ instead of ‘are more important’

Line 395: delete ‘not’ in ‘not to be’
Line 397: capital G needed on gripping
Line 401: a sentence should not begin with ‘and’, just continue this from the previous one.

Experimental design

The main ‘design’ elements in this study are 1) the behavioural sampling, of which I can’t judge because not enough detail has been provided, and 2) the statistical analysis. The analysis is complex hard to follow. I am not convinced that so many tests for significant differences were required for the purpose of this study. Whether the p-value is significant or not shouldn’t really matter, especially here, as this is a descriptive study documenting the types of manipulation strategies used by feeding elephants. The aim, to work out if there is more or less individual variation when elephants manipulate larger more ‘simple’ objects versus performing complex delicate tasks doesn’t surely need all these tests to tell us that?

It’s very difficult to judge statistics when you can’t see the data, and you’re new to the study, so I won’t try to tell you what should have been done, but it is hard to follow what was done and why here, and leaves me feeling there must have been a simpler way to analyse the data.

Please also use the guidelines set out by Budaev (https://onlinelibrary.wiley.com/doi/abs/10.1111/j.1439-0310.2010.01758.x) when describing what type of PCA was done and please report the relevant statistics (KMO & Bartlett’s test as well as a factor loading table).

The PCA and the PCA figures make sense, but you could help the reader and tie into your text by circling the three different groups from Figure 2 in different colours. Figure three is good also. However, figure 4 seems to me like it would contain all of the same information if the significance testing were removed. This would also allow you to include the rare behaviours, to show if they were performed most my certain individuals. I’m also not convinced that the two groups need to be treated separately, unless you are expecting group-effects.

Lines 200-206: in relation to my previous comments, I don’t feel like these differences needed to be tested for significance, it’s quite clear from Figure 2 what the relationship is between the different items. Keep this in if you wish to, but I’m not convinced it adds much.

Line 215: this sentence doesn’t make sense ‘whereas in group A elephants in the frequency of using it’

For the inter-individual differences section: I don’t see why the elephants were analysed according to group A and B, since you’re looking at individuals, they can all be compared to each other. And again, I am not convinced the significant differences really add anything; p-values are fairly meaningless in this context. If you disagree, this is fine, it’s not necessary to change it.

Validity of the findings

The findings all appear to be valid, but it should be acknowledged that no male elephants were included in the study, nor were any tusked elephants, which may impact handling strategies.

Additional comments

This manuscript is well written, but it could be improved by using an English Language service.

Reviewer 3 ·

Basic reporting

Thank you for submitting this manuscript. I really enjoyed reading it. My only substitutive comment is on the English language.

I found your interesting findings a little bit difficult to follow in places, which leads me to the recommendation that your English language should be improved to ensure that an international audience can clearly understand your text. This is partly due to some grammatical issues including the incorrect use of plurals (e.g. abstract lines 4 and 11 and main text lines 68, 95, 111, 212, 312) and tense changes both within sentences and within paragraphs (e.g. lines 100, 212-219, 238, 281, 368). For example: “The absence of bones in the trunk allows substantial degree of freedom for the movements in all directions” should be “The absence of bones in the trunk allows a substantial degree of freedom for movement in all directions”. These grammatical errors are extremely common errors for non-native speakers so you not alone in making these errors. It will also help the clarity and flow of your work if you try to stick to one tense, such as the past tense, when describing your results (e.g. lines 212-219). I also found some of your sentence structures a little convoluted in places and some of the discussion section in particular was a little long. This could be improved by writing in an active and precise scientific style. I’ve given a few examples here and below, but you do need to thoroughly check the whole manuscript. I am confident that you can improve this.

Minor comments:
- Your current title is a little bit vague as you just say elephant behaviour varies and not what type of behaviour. It would be great if you could be more specific as it will help your audience locate your article.
- Proboscideans (or even Elephantinae/ Elephants) are the largest terrestrial mammals - the blue whale is the largest mammal.
- Abstract: “impact of item properties” should be “food item” for clarity
- I am not entirely sure about your use of the word "personal" in your abstract as it is a slight anthropomorphism of your study subjects. Even in the field of animal personality, researchers tend to talk about inter-individual differences.
- Line 32: Use either "in addition" or "also" not both
- Line 64: "Evocated" - evaluated?
- Lines 37-64. I recommend you slightly re-structure this paragraph (or paragraphs - I'm not sure if there is a break in there) so that the paragraph is in a more logical order that creates a clear narrative. For example, you give specific details about the size of the trunk on lines 43-44 but the species these trunks belong to are not introduced with their latin names until lines 51-52. On lines 51-52, you say that African elephants "pinch" more than Asian elephants, but do not describe what that behaviour is until the following line. Providing the information required to interpret sentences before the current sentence will help your readers.
- Line 68: "expected" - hypothesised?
- Line 68: “the more simple is the task” – the simpler the task
- Line 80: Delete "are"
- Lines 82 & 83: "to" Beauval, should be "at"
- Lines 113-114: "soil" - do you mean flooring?
- Line 143: "this" should be "which"
- Line 274: "od" – of
- Lines 396-401: Please check these sentences as there seems to be a capital letter missing and odd placement of full stops.
- Figures 3 & 4: Please display the full error bar (not just the top half) as it will aid in interpretation.

Experimental design

The research questions and the methods, including Table 1, are generally well described, although the English needs some work as detailed above. I commend the authors on their detailed collation of trunk behaviours in this study.

Minor comments:
- Line 76: In what year?
- Table 1: Shake - "stir" implies a circular movement - is that what you mean or do you mean side-side or up and down.

Validity of the findings

The only word of caution I would give on your otherwise excellent results is that you have measured occurrences and not the duration of behaviours, which means a behaviour of a long duration and a short duration are given equal weight. This could impact your results, particularly when considering individual variability, and should be acknowledged in your discussion section.

The conclusion and the abstract could do with a little bit of work to clearly link back to the aims and hypotheses, and to summarise the results. For example, in the conclusion, it would be great if you could add a sentence or two about your findings related to the diameter of branches and size of the food item.

---

## Round 0.2 · Minor Revisions

A thorough revision in nearly all respects. The language, interpretation and emphasis issues have all been addressed. I accept that you are writing a second paper looking at lateralised responses.

However, this leaves the questions raised about the PCA. Your rebuttal letter states that details about the PCA have been added, but the manuscript with tracked changes does not show any substantial additions. I accept the point that not all of the details mentioned in the Budaev article (recommended by Reviewer 2) would be commonly added, but there are some key points in Budaev's article that are relevant.

Could you provide more information about any steps you took to avoid errors in the PCA arising from any issues with sample size or
case to variable ratio etc. Did you consider your sample size sufficient? Most importantly (and Budaev's final point) could you include the original correlation matrix as part of the supplementary materials, allowing re-analysis if necessary.

The reference on line 236 does not help, as I can only see raw data in the supplementary file and not the correlation matrix.

---

## Round 0.3 · accepted · Accept

Thanks for submitting the PCA information. Please however check line 190 which says that there are 10 columns/behaviours in the PCA table now provided in supplementary materials. I can see 12.